# Detecting Oxidative Stress Biomarkers in Neurodegenerative Disease Models and Patients

**DOI:** 10.3390/mps3040066

**Published:** 2020-09-24

**Authors:** Yulia Sidorova, Andrii Domanskyi

**Affiliations:** Institute of Biotechnology, HiLIFE, University of Helsinki, 00014 Helsinki, Finland

**Keywords:** neuroketals, nitrotyrosine, 8-hydroxy-2′-deoxyguanosine, malondialdehyde, immunohistochemistry, neurodereneration, oxidative stress, lipid peroxidation, reactive oxygen species (ROS)

## Abstract

Oxidative stress is prominent in many neurodegenerative diseases. Along with mitochondrial dysfunction and pathological protein aggregation, increased levels of reactive oxygen and nitrogen species, together with impaired antioxidant defense mechanisms, are frequently observed in Alzheimer’s, Parkinson’s, Huntington’s disease and amyotrophic lateral sclerosis. The presence of oxidative stress markers in patients’ plasma and cerebrospinal fluid may aid early disease diagnoses, as well as provide clues regarding the efficacy of experimental disease-modifying therapies in clinical trials. In preclinical animal models, the detection and localization of oxidatively damaged lipids, proteins and nucleic acids helps to identify most vulnerable neuronal populations and brain areas, and elucidate the molecular pathways and the timeline of pathology progression. Here, we describe the protocol for the detection of oxidative stress markers using immunohistochemistry on formaldehyde-fixed, paraffin-embedded tissue sections, applicable to the analysis of postmortem samples and tissues from animal models. In addition, we provide a simple method for the detection of malondialdehyde in tissue lysates and body fluids, which is useful for screening and the identification of tissues and structures in the nervous system which are most affected by oxidative stress.

## 1. Introduction

Oxidative stress is defined as an imbalance between the formation and destruction of free radicals [1]. The main source of free radicals in the body is oxygen activation occurring as a result of normal or pathological metabolic processes in cells. This leads to the formation of highly reactive superoxide anions (O2′^-^), hydrogen peroxide (H_2_O_2_) and singlet oxygen ^1^O_2_, called reactive oxygen species (ROS), which can interact with cellular molecules and damage them [2]. Lipid peroxidation is one of the major processes occurring in oxidative stress. During lipid peroxidation, ROS attack polyunsaturated fatty acids and induce intramolecular rearrangements of structure, with the formation of conjugated diene. In the next step, diene reacts with oxygen molecules, resulting in the generation of lipid radicals. The radicals further disintegrate with the formation of highly reactive 4-hydroxy-2-noneal (HNE), malondialdehyde (MDA) and acrolein (Figure 1A). These products can, in turn, react with, e.g., cellular proteins forming Shiff bases and Michael-adducts, thus damaging cellular macromolecules [1]. In the brain and retina, free radicals induce peroxidation of docosahexenoic acid (DHA), which is especially enriched in the synaptic membranes. DHA is oxidized to isoprostane-like compounds (Figure 1B), which undergo rearrangement to form D- and E-ring neuroprostanes, also called neuroketals, due to their specific enrichment in the nervous system (Figure 1B) [3,4]. Isoprostanes are prostaglandin-like compounds which are unique markers of oxidative stress, because they are formed only nonenzymatically in vivo in the process of free radical-initiated peroxidation [5].

In addition to lipids, ROS can also cause oxidative damage to proteins and nucleic acids, affecting essential cellular functions and activating cell death pathways [6]. For example, peroxynitrite (ONOO^−^), formed by the reaction of superoxide anion with NO, can oxidize lipoproteins and also trigger lipid peroxidation (via production of hydroxyl radical) with the formation of highly reactive products such as MDA [7]. In proteins, NO can nitrate tyrosine residues (Figure 1C) [8,9]. The resulting 3-nitrotyrosine can be relatively easily identified using immunological methods or chromatography [10], and therefore serves as a convenient oxidative stress biomarker. ROS can also attack and oxidize mitochondrial and nuclear DNA and RNA, with the guanine base being one of the most susceptible to oxidation [11,12]. Reaction with hydroxyl radicals leads initially to the generation of radical adduct. Then, 8-hydroxy-2′-deoxyguanosine (8-OHdG) is formed by one electron abstraction (Figure 1D); 8-OHdG exists in equilibrium with its tautomer 8-oxo-7,8-dihydro-2′-deoxyguanosine (8-oxodG), and in the literature, both names are used interchangeably [12,13]. Similarly to 3-nitrotyrosine, 8-OHdG is detected using immunological, chromatography or mass-spectrometric assays.

Due to high metabolic demands coupled with the active consumption of oxygen, a low level of antioxidant defense system components and saturation with polyunsaturated fatty acids, brain is especially vulnerable to lipid peroxidation and subsequent pathological processes [1,14,15,16]. Indeed, evidence of lipid peroxidation has been reported in many neurodegenerative disorders such as Parkinson’s disease (PD), Huntington’s Disease (HD), Alzheimer’s disease (AD) and amyotrophic lateral sclerosis (ALS).

### 1.1. Parkinson’s Disease

PD is diagnosed based on the appearance of motor symptoms such as resting tremor, slowness of movement, rigidity and dyskinesias, which are caused by the degeneration of nigrostriatal dopamine neurons [17]. It is the second most common neurodegenerative disorder with a prevalence of 0.4% in general population, and occurs more often in males and elderly people [18]. The exact reasons for particular vulnerability of the nigrostriatal dopamine system to degeneration are yet to be established, but such vulnerability may be attributed to the generation of free radicals in the process of the enzymatic or nonenzymatic degradation of dopamine associated with the production of ROS [19]. In pathophysiological analyses of postmortem samples, protein aggregates called Lewy bodies may be detected in the brain of the majority of PD patients [3,20,21]. Lewy bodies contain aggregated α-synuclein as a main component, as well as a number of other proteins and lipid membrane fragments. Aggregation of misfolded α-synuclein can impair axonal transport and proteostasis, leading to degeneration of neurites and cell death [22,23].

In the context of oxidative stress, elevated concentrations of MDA and NME have been reported in the plasma and cerebrospinal fluid (CSF) of PD patients [24]. Lipid peroxidation products such as acrolein have been shown to modify α-synuclein, resulting in disturbances in mitochondrial function [1,14]. Moreover, the HME and MDA adducts are detectable in Lewy bodies, implying the involvement of a lipid peroxidation process in PD pathogenesis [14].

### 1.2. Huntington’s Disease

HD is a hereditary disorder with a prevalence between of 2.17 to 7.33 cases per 100,000 people, depending on the geographic region [25]. In HD, an expansion (>36) of trinucleotide (CAG) sequence repeats leads to an abnormally long polyglutamine stretch in mutant huntingtin protein, and to the production of aggregation-prone homopolymeric proteins. Increasing length of the expanded CAG repeat sequence correlates with earlier disease onset [26,27,28]. Major symptoms of HD include severe motor and psychiatric disturbances caused by the degeneration of striatal GABAergic medium spiny neurons and the cortical neurons projecting to them. Progressive degeneration of these neurons results in the gradual worsening of symptoms and, eventually, death at about 15–20 years after disease onset.

Aggregation of mutant huntingtin and homopolymeric proteins leads to the appearance of nuclear and cytoplasmic inclusions in affected neuronal populations, causing proteostasis dysregulation, neurotransmission defects, mitochondrial dysfunction and oxidative stress [27,28,29,30,31]. Decreased capacity of antioxidant defense systems and increased oxidative stress contribute to neurodegeneration [29] and, indeed, the levels of oxidative stress markers, such as lipid peroxidation products and lactate in plasma, increase with the disease progression from the asymptomatic to symptomatic HD phase [32]. HD patients exhibit increased levels of 8-OHdG and nitrotyrosine in the brain, as well as in serum and leukocytes [33,34,35]. Importantly, 8-OHdG levels in leukocytes may serve a very sensitive biomarker of HD progression [29,36].

### 1.3. Alzheimer’s Disease

AD is currently an incurable neurodegenerative disorder manifesting as age-related progressive cognitive decline in affected patients [37]. AD is considered to be the cause of 50–75% of all dementia cases, and is estimated to affect more than 30 million people worldwide [38,39,40].

The exact causes of AD are not clear, though mutations in the *PSEN1, PSEN2* and *APP* genes and *APOE* gene variants can contribute to its development. AD is characterized by progressive synaptic loss and neurodegeneration affecting multiple brain regions [41,42,43,44]. The main hallmarks of pathology observed in AD brains are extracellular accumulation of amyloid-β (Aβ) peptides and hyperphosphorylation of the microtubule-associated protein Tau, leading to the accumulation of amyloid plaques and neurofibrillary tangles [42,45].

Oxidative stress has long been implicated in AD pathology. Complex interplay between proteostasis inhibition caused by abnormal protein aggregation, mitochondrial dysfunction, altered calcium dynamics, increased ROS production as well as depletion of cellular antioxidant defense mechanisms creates a vicious cycle of pathology progression leading to synaptic dysfunction, synaptic loss and eventual neurodegeneration [46,47,48,49]. On a molecular level, lipid peroxidation, nuclear and mitochondrial DNA damage and increased protein oxidation markers have been detected in the brain, CSF or plasma of AD patients at different stages of the disease [50,51,52,53,54,55,56,57,58].

### 1.4. Amyotrophic Lateral Sclerosis

ALS is a rare neurodegenerative disorder caused by the degeneration of motor neurons in the brain and spinal cord with a prevalence 4.1–8.4 per 100,000 people. The typical onset of the symptoms occurs at 51–66 years and the majority of patients die within 2–5 years after diagnosis [59,60]. The exact reason of motor neuron death in ALS is yet to be determined. The vast majority of ALS cases are sporadic, but approximately 10% are inheritable. Among familial ALS patients, about 20% carry mutations in superoxide dismutase (SOD) gene coding an enzyme of antioxidant defense destroying superoxide radicals in the body [61,62]. Mutant SOD can produce ROS via aberrant reaction, as shown in in vitro assays. Consistent with the ROS-producing capacity of mutated SOD, increased levels of protein adducts with HNE were found in animal ALS models. In particular, HNE modified dihydropyrimidinase-related protein 2 (DRP-2), heat shock protein 70 and α-enolase were reported to be present in the tissues of animals with ALS-like symptoms [1,14]. Thus, it is clear that ROS and lipid peroxidation may play a role in the pathogenesis of ALS.

### 1.5. Is Oxidative Stress the Cause or a Consequence of Pathological Changes in Brain Cells?

It is still unclear whether oxidative stress is the cause or a consequence of other pathological cellular changes, such as pathological protein aggregation, neurotransmitter oxidation, mitochondrial dysfunction, inflammation or deregulation of antioxidant pathways [29,46,63,64,65,66]. Nevertheless, reducing cellular ROS levels may be considered a promising neuroprotective or neurorestorative therapy, not only for the four neurodegenerative diseases mentioned above, but also for other conditions damaging the brain, for example, traumatic brain injury [67]. However, the application of antioxidant drugs for the treatment of neurodegenerative diseases has mostly been unsuccessful, probably, due to incorrect and/or insufficient therapy timing and their low penetrance through the blood-brain barrier [6,68,69]. Thus, further studies are needed to understand the role of oxidative stress in the pathogenesis of neurodegenerative disorders, to identify druggable targets involved in the oxidative stress-related processes and to develop novel therapeutics for the management of neurodegeneration.

## 2. Experimental Design

The overall level of oxidative damage in the organism can be evaluated by monitoring oxidative/nitrosative stress markers such as MDA in body fluids such as blood, serum, cerebrospinal fluid and even urine. However, we are generally still lacking the methods to reliably detect and monitor levels of oxidative damage in living tissue [70]. Nevertheless, considerable progress has been achieved in detecting oxidative stress markers in tissue lysates and histological sections. Many products formed in the process of lipid peroxidation can be measured using biochemical methods, but some of these measurements are expensive, as they require complicated purification processes, modern equipment and significant labor input. However, the evaluation of the content of MDA and other aldehydes produced during lipid peroxidation (or the level of thiobarbituric acid-reacting substances, TBARS) can be easily performed in a color reaction with thiobarbituric acid. Although the method has a number of limitations, the greatest of which is the fact that the readout reflects the amount of TBARs rather than MDA itself, it can still be useful for rapid screening and quantitative assessment of the intensity of lipid peroxidation processes in preclinical and clinical samples. Coupled with more sophisticated methods of oxidative stress analysis such as immunohistochemistry with specific antibodies against oxidative stress markers, it can provide insights into the mechanisms and pathogenesis of neurodegenerative disorders and facilitate the process of drug discovery. Here, we describe protocols to detect MDA in tissue lysates and oxidative stress markers in formaldehyde-fixed, paraffin-embedded brain sections. Together, these methods provide complementary information on oxidative stress levels. While the MDA detection in tissue samples can be suitable for clinical settings, it lacks the cellular resolution necessary to study oxidative stress levels in specific cell types. In contrast, the detection of oxidative stress markers in tissue sections utilizing standard imunohistochemical methods makes it possible to distinguish oxidative stress levels in individual cells and, by costaining with specific markers, in selected neuronal populations. For immunohistochemical detection, suitable tissue samples have to be obtained either by biopsy or postmortem, which is a clear drawback of the method. However, it is extremely useful for preclinical studies in animal models.

As an example of immunohistochemical detection, we describe here a protocol for detecting nitrotyrosine, neuroketals or 8-OHdG in dopaminergic neurons, visualized by costaining with antibody detecting tyrosine hydroxylase (TH), an established marker for this neuronal population.

### Materials

To prepare the buffers, either dd (double distilled) or milliQ H_2_O can be used.

10× Citra-buffer (BioGenex #HK086-9K)30% H_2_O_2_ (J.T.Baker #7047 or Merck #1.07209.1000)Normal swine serum (Dako #X0901)Antinitrotyrosine antibody (Alpha Diagnostic #NITT12-A) dilute 1:200Antineuroketals antibody (Chemicon #AB5611) dilute 1:2000Anti-8OHdG antibody (Chemicon #AB5830) dilute 1:1000Anti-TH antibody (Chemicon #AB1542) dilute 1:2000Biotinylated goat antirabbit IgG antibody (H+L) (Vector Laboratories #BA-1000) dilute 1:400Biotinylated horse antigoat IgG antibody (H+L) (Vector Laboratories #BA-9500) dilute 1:400Biotinylated rabbit antisheep IgG antibody (H+L) (Vector Laboratories #BA-9600) dilute 1:400Avidin-biotin-HRP (ABC) kit (Vector Laboratories # PK-4000)3, 3′-diaminobenzidine (DAB) (Sigma #D4293-50SET)HistoGreen kit (Linaris #E109)

## 3. Procedure

### 3.1. Immunohistochemical Detection of Oxidative Stress Markers in Dopaminergic Neurons

Below, we describe the detailed protocol for immunohistochemical detection of nitrotyrosine, neuroketals and 8-OHdG in a selected neuronal population, using dopaminergic neurons as an example. Dopaminergic neurons are readily visualized with antityrosine hydroxylase (TH) immunostaining. Other markers (such as dopamine transporter (DAT), acetylcholine esterase (AChE), neuronal nuclei (NeuN), β-III-tubulin etc.) can be used to study different populations of neural cells [71].

#### 3.1.1. Deparaffination and Rehydration. Time for Completion: Approx.60 min

During this step, traces of paraffin are removed and the sections are rehydrated. It is practical to place the slides in a slide rack and move the rack between containers with corresponding solutions (i.e., three containers with xylene, six with different ethanol concentrations, and one with PBS).


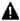
**CRITICAL STEP** It is important to not let the slides dry at any stage of the protocol.

Place the slides in a slide rack and incubate for 10 min in xylene
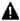
**CRITICAL STEP** Work with xylene must be done in a fume hood.Repeat twice with fresh changes of xylene.Incubate twice for 5 min in absolute ethanol.Incubate twice for 5 min in 95% ethanol (v/v solution in H_2_O).Incubate twice for 5 min in 70% ethanol.Incubate 10 min in PBS.

#### 3.1.2. Antigen Retrieval and Peroxidase Blocking. Time for Completion: 1 h 30 min

Heat-induced antigen retrieval improves the binding of the antibody by improving the accessibility of the target epitopes, which can be masked during the fixation and paraffin embedding process. For the detection of antigen-bound antibodies, horseradish peroxidase (HRP) enzyme is utilized; its localization on the sections is detected by the presence of an insoluble colored product of HRP-catalyzed conversion of chromogenic substrate (DAB or HistoGreen). If not blocked, the endogenous peroxidase will also convert the chromogenic substrate, resulting in unspecific background staining. Incubation with saturating amounts of H_2_O_2_ efficiently and irreversibly inactivates the endogenous peroxidase. While most protocols suggest blocking peroxidase in 3% H_2_O_2_ solution [72], such a high concentration of peroxide may affect the epitopes on some antigens recognized by antibodies. We have empirically found that for the antibodies used in this protocol, blocking in 1.5% H_2_O_2_ solution produces optimal results with reasonably low background.

Place the slides in heat-resistant container with sufficient 1× Citra-buffer to completely cover them and heat in a microwave oven for 3 min at 800 Watt.
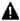
**CRITICAL STEP** Check that the solution is boiling.Alternatively, antigen retrieval can be performed in 0.1 M citrate buffer, pH 6.0: 40 mM sodium citrate dihydrate, 60 mM citric acid.Continue heating for an additional 8 min at 360 Watt.Let the container with the slides cool at room temperature for 20–30 min.Wash the slides three times for 5 min in PBS.Block endogenous peroxidase by incubating the slides 30 min in 1.5% H_2_O_2_ solution.Wash the slides three times for 5 min in PBS.

#### 3.1.3. Immunolabelling with the First Antibody. Time for Completion: 16–17 h

Before the sections are incubated with primary antibodies, the unspecific binding of proteins to tissue is blocked by incubation with 5% normal swine serum.

Remove liquid from the slides and carefully encircle the sections with a PAP pen (Merck #Z672548 or similar).
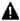
**CRITICAL STEP** The PAP pen will create a hydrophobic barrier so that the antibody solution will stay on the section. It is critically important that the hydrophobic solution does not come into contact with the section, as it will essentially prevent any aqueous solution from reaching the section.Place each slide horizontally in a moist chamber and cover each brain section with approximately 100 µL of blocking solution.
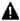
**CRITICAL STEP** It is critically important for the blocking solution at this step, and for the antibody solution in the following steps, to cover the entire section; therefore, the volume should be adjusted depending on the section size. Incubation in a moist chamber—e.g., a box lined with wet filter paper—prevents the evaporation of the solution, making it possible to reduce the solution volume.Incubate 30–60 min at room temperature.Carefully remove the blocking solution. Do not rinse the slides.Cover each brain section with approximately 100 µL of one of the primary antibodies diluted in blocking solution: antinitrotyrosine diluted 1:200, antineuroketals diluted 1:2000 or anti-8-OHdG diluted 1:1000.
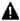
**CRITICAL STEP** Optimal antibody dilution should be empirically tested for each antibody and, ideally, for each new batch, in a pilot experiment. Dilutions may range from 1:50 up to 1:5000 and even higher, depending on the antibody and antigen.Incubate overnight (approximately 15 h) at 4 °C in a moist chamber.On the next day, move the slides to a slide rack and wash three times for 10 min in PBS.Remove liquid from slides, place them horizontally and cover each brain section with approximately 100 µL of the proper secondary biotinylated antibody diluted 1:400 in PBS.
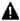
**CRITICAL STEP** The secondary antibody depends on the animal species in which the primary antibody was raised. For example, antinitrotyrosine antibodies (raised in rabbit) use antirabbit secondary antibodies, while antineuroketals or anti-8-OHdG antibodies (both raised in goat) use antigoat secondary antibodies.Incubate for 30 min at room temperature; meanwhile prepare the avidin-biotin-HRP (ABC) mixture.Move the slides to a slide rack and wash three times for 10 min in PBS.Remove the liquid from the slides, place them horizontally and cover each brain section with approximately 100 µL ABC mixture.Incubate at room temperature for 30 min.Move the slides to a slide rack and wash three times for 10 min in PBS.Remove the liquid from the slides, place them horizontally and cover each brain section with approximately 100 µL of DAB solution.Incubate at room temperature for 30 sec–10 min (depending on the antibody) until a brown color develops.
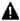
**CRITICAL STEP** The exact incubation time depends on the antibody and quality of the sections. For all slides in one staining batch, the same incubation time should be maintained to enable comparisons to be made of the staining between different slides. The intensity of DAB staining, however, is not linearly dependent on the antigen concentration [73].Stop the reaction by immersing the slides in H_2_O.Move the slides to a slide rack and wash three times for 10 min in PBS.

#### 3.1.4. Immunolabelling with the Second antibody. Time for Completion: 19–20 h

Block HRP from the first staining by incubating the slides for 30 min in 1.5% H_2_O_2_ solution.Wash three times for 10 min in PBS.Remove the liquid from the slides, place them horizontally and cover each brain section with several drops of avidin solution (Vector Laboratories #SP-2001).
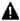
**CRITICAL STEP** Incubation with avidin and, subsequently, with biotin is necessary to block the nonspecific binding of avidin-conjugated HRP to the first secondary antibody.Incubate for 15 min at room temperature.Wash for 10 min in PBS.Remove the liquid from the slides, place them horizontally and cover each brain section with several drops of biotin solution (Vector Laboratories #SP-2001).
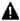
**CRITICAL STEP** Incubation with avidin and, subsequently, with biotin is necessary to block the nonspecific binding of avidin-conjugated HRP to the first secondary antibody.Incubate for 15 min at room temperature.Wash three times for 10 min in PBS.Cover each brain section with approximately 100 µL of one of the second primary antibodies diluted in blocking solution: antityrosine hydroxylase diluted 1:2000Incubate overnight (approximately 15 h) at 4 °C in a moist chamber.On the next day, move the slides to a slide rack and wash three times for 10 min in PBS.Remove the liquid from the slides, place them horizontally and cover each brain section with approximately 100 µL of proper secondary biotinylated antibodies diluted 1:400 in PBS.
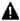
**CRITICAL STEP** The choice of secondary antibody depends on the animal species in which the primary antibodies were raised. For example, for antityrosine hydroxylase antibodies (raised in sheep), use antisheep secondary antibodies.Incubate for 30 min at room temperature; meanwhile prepare the avidin-biotin-HRP (ABC) mixture.Move the slides to a slide rack and wash three times for 10 min in PBS.Remove the liquid from the slides, place them horizontally and cover each brain section with approximately 100 µL ABC mixture.Incubate at room temperature for 30 min.Move the slides to a slide rack and wash three times for 10 min in PBS.Remove the liquid from the slides, place them horizontally and cover each brain section with approximately 100 µL of HistoGreen solution.Incubate at room temperature for 30 sec–10 min (depending on the antibody) until a green/blue color develops.
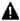
**CRITICAL STEP** The exact incubation time depends on the antibody and quality of the sections. For all slides in one staining batch, the same incubation time should be maintained to enable comparisons to be made of the staining between different slides.Stop the reaction by immersing the slides in ddH_2_OMove the slides to a slide rack and wash three times for 10 min in ddH_2_O.

#### 3.1.5. Dehydration and Mounting. Time for Completion: Approx. 15 h

For prolonged storage (i.e., years), the sections are dehydrated and covered with coverslips in a hydrophobic mounting medium.

Incubate twice for 3 min in 70% ethanol.Incubate twice for 3 min in 95% ethanol.Incubate twice for 5 min in absolute ethanol.Incubate twice for 5 min in xylene.
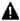
**CRITICAL STEP** Work with xylene must be done under a fume hood.Using a glass rod, put several drops of Eukitt mounting medium (Merck #03989 or similar) in each section.Cover the sections with coverslip to avoid the formation of bubbles.Keep overnight at room temperature under the fume hood to dry.

### 3.2. Evaluation of MDA Content in Tissue Lysates. Time for Completion Approximately 1 h

An easy technique to evaluate the level of lipid peroxidation in tissue samples is the analysis of the concentration of colored/fluorescent product formed in reaction with thiobarbituric acid in acidic conditions upon heating; this approach is widely referred to as MDA content measurement. The method is simple and straightforward, and the basic protocol is provided below, although numerous variations exist.

Homogenize the tissue sample in 50 mM Tris-HCl (pH 7.2) (10% *w/v* ratio).**OPTIONAL STEP** Cells can be fractionated before measurements by centrifugation if desired. Buffers and solutions other than Tris-HCl (pH 7.2) can be used to homogenize the tissue sample. In particular, Oakes et al. (2002) report using 1.15% KCl solution for this purpose [74]. Buffers containing tris-maleate (≤40 mM), imidazole (≤20 mM) or 4-morpholine propanesulfonic acid ≤ 20 mM) also do not interfere with the assay readout. However, avoid sucrose-containing buffers, as sucrose can interact with thiobarbituric acid [75].There is a concern regarding the continuous production of MDA during the process of sample handling. While it is generally not a big problem for preclinical measurements, where appropriate controls are easily available and results are usually interpreted within one experiment, it can be an issue with clinical samples. To minimize MDA production during sample handling, an antioxidant, e.g., butylated hydroxytoluene (35 µM), can be added to the tissue homogenization buffer [74]. To improve the solubilization of tissue components, sodium dodecyl sulfate can be added to the reaction mixture at concentrations of, e.g., 12.4 mM [74,75].Add trichloroacetic acid (7–10% of the final volume) and mix well. We successfully used the following ratio: 700 µL of homogenate and 500 µL of 15% trichloroacetic acid.Centrifuge at 10,000× *g* for 10 min to remove precipitated proteins.Collect the supernatant and mix it with 0.75% thiobarbituric acid (1:1), incubate in a water bath at 100 °C for 15–20 min, cool on ice for 5 min and centrifuge at 1000× *g* for 10 min to remove any insoluble inclusions.
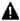
**CRITICAL STEP** Thiobarbituric acid is poorly soluble in water; its aqueous solution is unstable and should be prepared freshly before each test. Some manuals recommend dissolving thiobarbituric acid directly in 20% trichloroacetic acid to a 0.5% final concentration. According to these manuals, such a solution can be stored for about a month at room temperature without precipitation.Measure the absorbance of the resulting supernatant at 532 nm (A532), with 580 nm as the reference wavelength (used for baseline correction) using a suitable spectrophotometer with a cuvette with a width of 1 cm.**OPTIONAL STEP** It is also possible to measure the assay readout fluorometrically using the following settings: excitation at 515 nm, emission at 553 nm [74].MDA quantity is calculated as C(MDA) = (A532 − A580)/K, where K is a millimolar extinction coefficient equal to 155 mM^−1^ cm^−1^. Blanc sample containing the sample homogenization buffer and undergoing the same treatment as experimental samples on all steps of the analysis should be included [76,77].**OPTIONAL STEP** The reaction product can be extracted with n-butanol to assess the quantity of the reaction product only in lipid fraction [74,75].

## 4. Expected Results

The reliable interpretation of the immunostaining results strongly depends on the quality of the histological sections. Here, we describe the protocol for the immunostaining of histological sections from formaldehyde-fixed brains embedded in paraffin. Meanwhile, in the present study, fixation in formaldehyde, paraffin embedding and sectioning at 5 µm thickness was shown to be a robust method to obtain good quality brain sections for immunostaining. This method requires more hands-on time than producing free-floating 30–40 µm sections on vibratome or cryostat. Also, using fluorescently labelled secondary antibodies and confocal microscopy makes it possible to more precisely colocalize antigens. However, the fluorescently labelled sections cannot be stored for prolonged periods due to the fading that occurs of the fluorescent signal. In contrast, immunohistochemically stained sections can be stored for years without noticeable loss of signal, and can easily be scanned in an automated slide scanner (such as Pannoramic Digital Slide Scanners (3DHistech)) and used for unbiased computer-assisted image quantification [78]. Representative images of immunostained sections are shown in Figure 2; see also [79].

Although easy to perform, the present method for the evaluation of MDA content in tissue lysates has a number of limitations which should be kept in mind when interpreting the results. First of all, not only MDA, but also many other aldehydes produced during the lipid peroxidation process, and lipid-peroxidation unrelated substances, such as some proteins, sucrose, urea, plasma components, etc. may react with thiobarbituric acid and form a colored product, leading to overestimations of the MDA concentration in the sample. The presence of blood in the sample, in particular, hemolyzed erythrocytes, can interfere with the assay readout. To minimize interference from sialic acid, correcting absorbance values collected at wavelength λ_532_ to values collected with λ_/__580_ is recommended [75].

To conclude, the analysis of TBARS concentration in tissue samples is a cheap and easy screening assay which can be performed prior to application of more advanced methods to analyze the products of oxidative stress in preclinical and clinical samples. The spatial resolution of the MDA analysis method to detect oxidative stress can be improved if it is followed by immunohistochemical analysis of tissues with high levels of TBARS for the presence of 8-OHdG, nitrotyrosine or neuroketals, as described above. The combination of these two techniques represents an affordable yet reliable option for analysis of oxidative stress levels in tissue samples.

## 5. Reagents Setup

Phosphate-buffered saline (PBS) pH 7.4: 137 mM NaCl, 2.7 mM KCl, 8 mM Na_2_PO_4_ and 1.5 mM KH_2_PO_4_ in H_2_O1× Citra-buffer: dilute 1 part of 10× stock solution (BioGenex #HK086-9K, stored at 4 °C) with 9 parts of H_2_O; prepare fresh before use1.5% H_2_O_2_ in PBS: dilute 12.5 mL 30% H_2_O_2_ (J.T.Baker #7047 or Merck #1.07209.1000) in 237.5 mL PBSBlocking solution: 5% (v/v) normal swine serum (Dako #X0901) in PBSAvidin-biotin-HRP (ABC) mixture (Vector Laboratories # PK-4000): to 1 mL PBS add 10 µL solution A, mix and add 10 µL solution B, mix, and let stand at room temperature for at least 20 min before use.15% Tricholoacetic acid0.75% Thiobarbituric acid (make fresh before each use): mix 0.75 g of thiobarbituric acid powder with 100 mL ddH_2_O and warm the solution until dissolved.50 mM Tris HCl buffer with pH 7.2–7.4 (1 L): Disslove 6.2 g Tris in 900 mL ddH_2_O, adjust pH with HCl solution and afterwards, adjust volume to 1 L.

## Figures and Tables

**Figure 1 mps-03-00066-f001:**
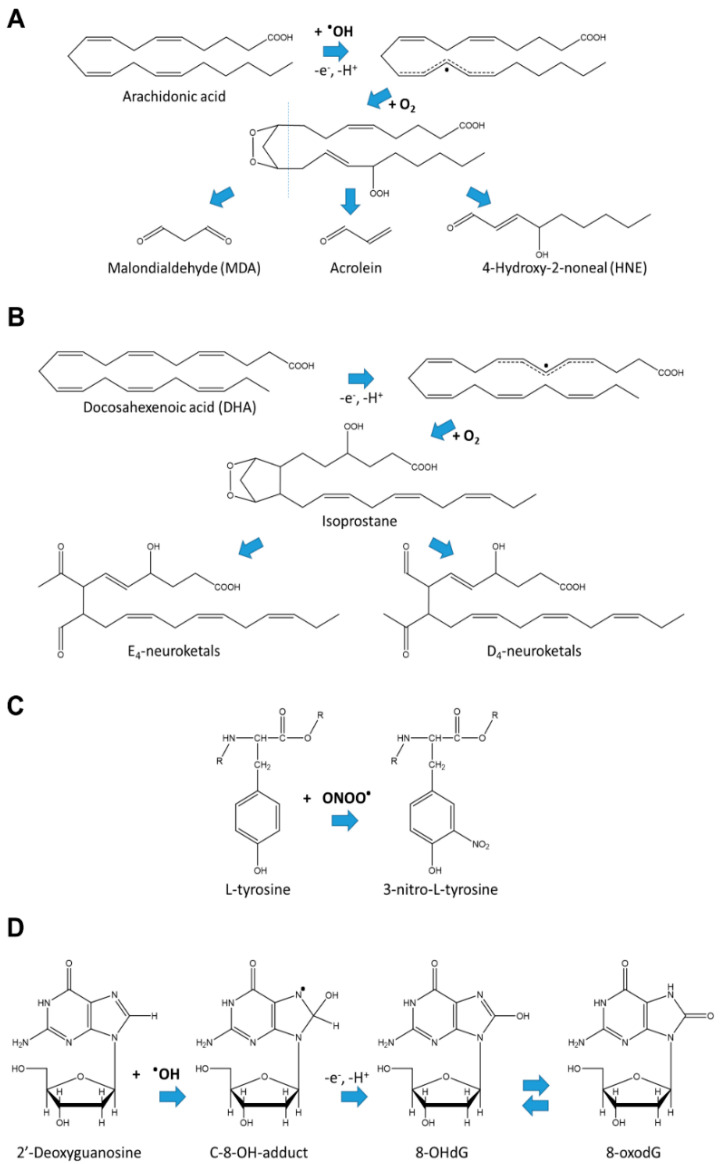
Example chemical reactions leading to formation of (**A**) lipid peroxidation products, (**B**) neuroketals, (**C**) nitrosylated tyrosine, and (**D**) 8-hydroxy-2′-deoxyguanosine (8-OHdG) in equilibrium with its tautomer 8-oxo-7,8-dihydro-2′-deoxyguanosine (8-oxodG).

**Figure 2 mps-03-00066-f002:**
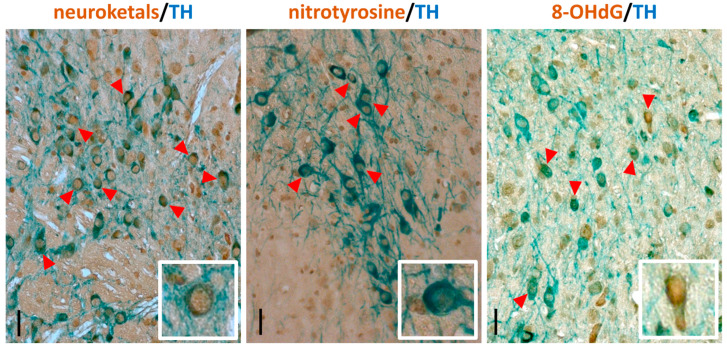
Representative images of mouse midbrain sections immunostained for 8-hydroxy-2′-deoxyguanosine (8-OHdG), nitrotyrosine and neuroketals (brown) in combination with tyrosine hydroxylase (TH, blue) to identify dopaminergic neurons. Scale bars: main image, 100 µm; inset, 40 µm.

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
