# Peer review of "Detecting Oxidative Stress Biomarkers in Neurodegenerative Disease Models and Patients"

_mps, 2020, doi:10.3390/mps3040066_

Round 1
Reviewer 1 Report
This is a well designed and accurate report, expected to meet the readers' interest.
A number of minor English mistakes are noted and corrected in the enclosed file.
In view of forthcoming studies, the authors are advised to better focus on mitochondrial dysfunction, which has paramount relevance in the pathogenesis of the disorders investigated in this paper.

Author Response
Reviewer 1:
This is a well designed and accurate report, expected to meet the readers' interest. A number of minor English mistakes are noted and corrected in the enclosed file. In view of forthcoming studies, the authors are advised to better focus on mitochondrial dysfunction, which has paramount relevance in the pathogenesis of the disorders investigated in this paper.
Thank you very much for positive evaluation of our work and for text the corrections. We agree with most of your comments; however, we would like to keep spelling out “oxidative stress” (not introducing OS abbreviation) throughout the manuscript. As we already have many abbreviations, we believe that introducing yet another one would make reading the text somewhat more difficult. Also, please, not that “Pannoramic Scanner” (with double “n”) is the name of the device.
We fully agree about the importance of mitochondrial dysfunction, as well as protein aggregation, for the pathogenesis of neurodegenerative diseases. However, in this manuscript we decided to focus on oxidative stress methods, also for the sake of its length; we are planning to cover methods to study mitochondrial dysfunction in a follow up manuscript.
Reviewer 2 Report
Review of the manuscript “Detecting oxidative stress biomarkers in neurodegenerative disease models and patients” by Yulia Sidorova and Andrii Domansky submitted to “Methods and protocols”, MDPI
Oxidative stress is a common biochemical event associated with many diseases, which can be defined as imbalance between production and elimination of free radicals. Disturbances in the normal redox state of cells can cause various toxic effects due to the production of peroxides and free radicals that impair various components of the cell, including proteins, DNA and lipids. It is important to know molecular details of the oxidative stress in order to better understand mechanisms of human diseases. The authors describe protocols for detection of oxidative stress markers using methods of immunohistochemistry. This is important field of investigation and the results presented in the manuscript will be interesting for the readers of the journal “Methods and protocols”, MDPI.
The manuscript is well written and contain new thought-provoking data and detailed descriptions of methods for researchers working in biochemistry and pathology, as well as for physicians.
The following corrections should be done.
Abstract
Line 20: ”applicable to the analysis of post-mortem samples and animal models” should be written as follows :” applicable to the analysis of postmortem samples and tissues from animal models”
Introduction
Lines 33-37: ”During lipid peroxidation ROS attack polyunsaturated fatty acids and induce intramolecular rearrangement of structure with formation of conjugated diene followed by its reaction with oxygen molecule resulting in generation of lipid radicals which disintegrate with formation of highly reactive 4-hydroxy-2-noneal (HNE), malondialdehyde (MDA) and acrolein”.
The sentence is too long and hard to read. It should be split into three sentences to become more reader friendly: “During lipid peroxidation ROS attack polyunsaturated fatty acids and induce intramolecular rearrangement with formation of conjugated diene. On the next step diene react with oxygen molecule resulting in generation of lipid radicals. The radicals further disintegrate with formation of highly reactive 4-hydroxy-2-noneal (HNE), malondialdehyde (MDA) and acrolein”.
Line 45 : “can oxidize lipoproteins and nitrate tyrosine residues (Figure 1C)…” The sentence sounds weird. Why the authors combine lipoproteins and nitrate tyrosine residues here. Do they mean tyrosine residues in other classes of proteins?
Line 131 After the paragraph describing Amyotrophic lateral sclerosis the authors make an intermediate conclusion beginning on line 133. It can be entitled “Is oxidative stress the cause or consequence of pathological changes in brain cells?”
Lines 136-137: ”Nevertheless, reducing cellular ROS levels seemed like a reasonable neuroprotective or neurorestorative therapy.” This is a clumsy sentence. Should be rewritten as follows: ”Nevertheless, reducing cellular ROS levels may be considered as promising neuroprotective or neurorestorative therapy not only for four neurodegenerative diseases mentioned above, but also for other conditions damaging the brain, for example, traumatic brain injury” (references should be added here:”Role of synucleins in traumatic brain injury an experimental in vitro and in vivo study. Mol Cell Neurosci 2014, 63, 114–23).
Line 139:” incorrect and/or insufficient therapy timing and low brain exposure…” the authors should explain what they mean by “low brain exposure”
Lines 144—145: “While we are still lacking the methods to reliably detect and monitor levels of oxidative damage in living tissue”. The authors should briefly mention here that the level of oxidative damage in living tissue may be detected indirectly by changes in blood, serum, CSF, etc.
Overall, the manuscript is well written, contains thoroughly described methods and will be helpful for many biochemists, pathologists and other researchers.
Author Response
Reviewer 2:
Review of the manuscript “Detecting oxidative stress biomarkers in neurodegenerative disease models and patients” by Yulia Sidorova and Andrii Domansky submitted to “Methods and protocols”, MDPI.
Oxidative stress is a common biochemical event associated with many diseases, which can be defined as imbalance between production and elimination of free radicals. Disturbances in the normal redox state of cells can cause various toxic effects due to the production of peroxides and free radicals that impair various components of the cell, including proteins, DNA and lipids. It is important to know molecular details of the oxidative stress in order to better understand mechanisms of human diseases. The authors describe protocols for detection of oxidative stress markers using methods of immunohistochemistry. This is important field of investigation and the results presented in the manuscript will be interesting for the readers of the journal “Methods and protocols”, MDPI.
The manuscript is well written and contain new thought-provoking data and detailed descriptions of methods for researchers working in biochemistry and pathology, as well as for physicians.
We thank the reviewer for positive evaluation and careful proofreading of our manuscipt, as well as for valuable comments.
The following corrections should be done.
Abstract
Line 20: ”applicable to the analysis of post-mortem samples and animal models” should be written as follows :” applicable to the analysis of postmortem samples and tissues from animal models”
Changed as requested
Introduction
Lines 33-37: ”During lipid peroxidation ROS attack polyunsaturated fatty acids and induce intramolecular rearrangement of structure with formation of conjugated diene followed by its reaction with oxygen molecule resulting in generation of lipid radicals which disintegrate with formation of highly reactive 4-hydroxy-2-noneal (HNE), malondialdehyde (MDA) and acrolein”.
The sentence is too long and hard to read. It should be split into three sentences to become more reader friendly: “During lipid peroxidation ROS attack polyunsaturated fatty acids and induce intramolecular rearrangement with formation of conjugated diene. On the next step diene react with oxygen molecule resulting in generation of lipid radicals. The radicals further disintegrate with formation of highly reactive 4-hydroxy-2-noneal (HNE), malondialdehyde (MDA) and acrolein”.
The sentence has been split according to the suggestion.
Line 45 : “can oxidize lipoproteins and nitrate tyrosine residues (Figure 1C)…” The sentence sounds weird. Why the authors combine lipoproteins and nitrate tyrosine residues here. Do they mean tyrosine residues in other classes of proteins?
We have split the sentence and further explained our line of thinking. In this particular case, we mean tyrosine residues both in lipoproteins and other classes of proteins as well.
Line 131 After the paragraph describing Amyotrophic lateral sclerosis the authors make an intermediate conclusion beginning on line 133. It can be entitled “Is oxidative stress the cause or consequence of pathological changes in brain cells?”
Thank you very much for this very useful suggestion. We have introduced this sub-heading.
Lines 136-137: ”Nevertheless, reducing cellular ROS levels seemed like a reasonable neuroprotective or neurorestorative therapy.” This is a clumsy sentence. Should be rewritten as follows: ”Nevertheless, reducing cellular ROS levels may be considered as promising neuroprotective or neurorestorative therapy not only for four neurodegenerative diseases mentioned above, but also for other conditions damaging the brain, for example, traumatic brain injury” (references should be added here:”Role of synucleins in traumatic brain injury an experimental in vitro and in vivo study. Mol Cell Neurosci 2014, 63, 114–23).
We have rewritten the sentence as suggested and introduced the required reference.
Line 139:” incorrect and/or insufficient therapy timing and low brain exposure…” the authors should explain what they mean by “low brain exposure”
We are sorry for misunderstanding; by low brain exposure we mean that corresponding drug might have low penetration through the blood-brain barrier which results in low brain concentration of the compound in the brain. This will generally be accompanied by low values of area under the pharmacokinetic curve “concentration-time”). In pharmacological literature this is generally referred to as brain exposure. However, we understand that majority of audience of the journal has different background and order to avoid unfamiliar term, we have changed the sentence accordingly.
Lines 144—145: “While we are still lacking the methods to reliably detect and monitor levels of oxidative damage in living tissue”. The authors should briefly mention here that the level of oxidative damage in living tissue may be detected indirectly by changes in blood, serum, CSF, etc.
We have added this information, as requested.
Reviewer 3 Report
Introduction: When authors refer to lipid peroxidation, they mention a marker (MDA) that os particularly produced following nitrosative stress. That is to be hioghlighted. Additionally, the isoprostane is mentioned but not fully explained.
Most of paragraphs of experimental section are merely elencation of reagents and procedures. This is unusual. Nevertheless, it could be considered after a revision of punctuation.
The resolution of the figure 2 should be increased.
Authors contributions section is not complete.
Author Response
Reviewer 3:
Introduction: When authors refer to lipid peroxidation, they mention a marker (MDA) that os particularly produced following nitrosative stress. That is to be hioghlighted. Additionally, the isoprostane is mentioned but not fully explained.
We have added a reference highlighting the importance of MDA as a lipid peroxidation marker. Furhter, we added a reference and a sentence explaining the chemical nature of isoprostanes, which is also illustrated on Figure 1B.
Most of paragraphs of experimental section are merely elencation of reagents and procedures. This is unusual. Nevertheless, it could be considered after a revision of punctuation.
We would like to point out that the manuscript is describing experimental protocols and, as such, is formatted according to the journal requirements. We have carefully checked punctuation and grammar according to reviewers’ comments.
The resolution of the figure 2 should be increased.
Please, note that the resolution of Figure 2 is 300 DPI, as required by the journal. However, if the reviewer meant increased magnification, we have added to Figure 2 insets showing higher magnification of example immunostained cells.
Authors contributions section is not complete
The paragraph stating author contributions can be found right after the section 5. We added some clarifications on the role of authors in the data collection and analysis and providing experimental evidences.